# Research Progress on Detection of Pathogens in Medical Wastewater by Electrochemical Biosensors

**DOI:** 10.3390/molecules29153534

**Published:** 2024-07-27

**Authors:** Bangyao Chen, Jiahuan He, Kewei Tian, Jie Qu, Lihui Hong, Qin Lin, Keda Yang, Lei Ma, Xiaoling Xu

**Affiliations:** 1Key Laboratory of Artificial Organs and Computational Medicine in Zhejiang Province, Shulan International Medical College, Zhejiang Shuren University, Hangzhou 310015, China; chenbangyao@outlook.com (B.C.); 202001020215@stu.zjsru.edu.cn (J.H.); weiketiantkw@163.com (K.T.); helloqj2024@163.com (J.Q.); 601175@zjsru.edu.cn (L.H.); hopeqinlin@foxmail.com (Q.L.); 2Beijing Key Laboratory of Fuels Cleaning and Advanced Catalytic Emission Reduction Technology, Beijing College of New Materials and Chemical Engineering, Institute of Petrochemical Technology, Beijing 102617, China

**Keywords:** electrochemical biosensors, medical wastewater, pathogen detection, public health, environmental safety, rapid detection

## Abstract

The detection of pathogens in medical wastewater is crucial due to the high content of pathogenic microorganisms that pose significant risks to public health and the environment. Medical wastewater, which includes waste from infectious disease and tuberculosis facilities, as well as comprehensive medical institutions, contains a variety of pathogens such as bacteria, viruses, fungi, and parasites. Traditional detection methods like nucleic acid detection and immunological assays, while effective, are often time-consuming, expensive, and not suitable for rapid detection in underdeveloped areas. Electrochemical biosensors offer a promising alternative with advantages including simplicity, rapid response, portability, and low cost. This paper reviews the sources of pathogens in medical wastewater, highlighting specific bacteria (e.g., *E. coli*, *Salmonella*, *Staphylococcus aureus*), viruses (e.g., enterovirus, respiratory viruses, hepatitis virus), parasites, and fungi. It also discusses various electrochemical biosensing techniques such as voltammetry, conductometry, impedance, photoelectrochemical, and electrochemiluminescent biosensors. These technologies facilitate the rapid, sensitive, and specific detection of pathogens, thereby supporting public health and environmental safety. Future research may should pay more attention on enhancing sensor sensitivity and specificity, developing portable and cost-effective devices, and innovating detection methods for diverse pathogens to improve public health protection and environmental monitoring.

## 1. Introduction

Wastewater in the medical system contains many pathogenic microorganisms, which refer to microorganisms that can invade humans and animals and cause infections or even infectious diseases, including viruses [1,2,3], bacteria [4,5], fungi [6,7], parasites [8], etc., and are one of the biological factors affecting biosecurity. The composition of medical wastewater is complex [9] and can be divided into two categories: wastewater from infectious diseases and tuberculosis medical institutions, and wastewater from comprehensive medical institutions and other medical institutions, containing physical and chemical pollutants, pathogenic microorganisms, radioactive pollutants, etc. Among them, in high-risk conditions of cross-infection such as medical institutions, unhygienic toilet conditions and unsafe emissions of pollutants such as faeces cause 1.5 million people worldwide to become infected with intestinal diseases and are one of the leading causes of child death in less-developed countries and regions [10]. In addition, according to statistics [11], more than one-third of the wastewater produced by hospitals is dangerous infectious waste, and pathogenic bacteria such as *Shigella Castellani*, *Salmonella Typhimurium*, *Vibrio cholerae*, *Mycobacterium tuberculosis*, *Salmonella*, *Brucella* and Enterovirus, Poliovirus, Coxsackie virus, Ecovirus, Adenovirus, Hepatitis virus, and so on can be detected in medical wastewater. An experimental team [12] used 16S rRNA gene fingerprinting to estimate the abundance of the total population of potential pathogenic bacteria in Hong Kong municipal wastewater, which was found to be 0.06–3.20%. These pathogens can cause diarrhoea, acute gastroenteritis, sepsis, and other diseases, the incidence of which is high [13]. In addition, the gut microbiome in these faeces is directly linked to human health, influencing the development of chronic diseases ranging from metabolic diseases [14] to gastrointestinal disorders [15] and colorectal cancer [16,17]. Improper disposal of infectious hospital wastes and wastewater can infect humans through contaminated food or water and cause great harm to public health and the ecological environment [18,19,20]. Hence, it is crucial to conduct research on rapid detection methods for pathogenic microorganisms in medical wastewater.

At present, the detection methods of pathogenic microorganisms in medical wastewater mainly include nucleic acid detection [21,22], immunological detection [23,24,25], and electrochemical biosensor detection [26]. Among them, nucleic acid detection is widely used in the fields of disease diagnosis, gene expression and biological identification, and molecular biological testing technologies such as polymerase chain reaction (PCR) and loop-mediated isothermal amplification (LAMP). Since nucleic acid is usually quantitative, nucleic acid-based detection methods generally need to expand the content of the target sequence by effectively amplifying it, that is, nucleic acid amplification technology. However, nucleic acid amplification methods require expensive test platforms and need to design suitable primers, and the process is cumbersome and time-consuming, which is not suitable for rapid detection of pathogenic microorganisms in faeces in underdeveloped areas lacking relevant equipment [27,28]. The rapid detection methods based on immunology, such as rapid immunological assays like the enzyme immunoassay (EIA) and enzyme-linked immunosorbent assay (EIASA) [25,29], use specific antigen-antibody reaction to conduct quantitative and qualitative detection of the target object according to the antigen or antibody of the target object. However, these methods have disadvantages such as their time-consuming nature, cross-reactions, and false negatives. Electrochemical biosensors have the advantages of simple operation, short reaction time, portability, miniaturization, low cost, etc., and can convert the reaction between biomarkers and detection elements into electrical signals, so that the detection signal is easier to read, and they can reduce the impact of background values, so that the detection results are more accurate [30,31]. Electrochemical biosensor technology includes volt-ampere type biosensors, conductive biosensors, impedance biosensors, photoelectric chemical biosensors, and electrochemical luminescence biosensors. In this paper, pathogenic microorganisms and corresponding electrochemical biosensor detection methods are mainly discussed in order to provide reference and help for the development of pathogenic microorganisms’ detection methods in medical wastewater.

## 2. Pathogens in Medical Wastewater

There is a high probability that the wastewater of medical institutions contains pathogenic microorganisms, and if there is no sewage treatment facility or disinfection is incomplete, it will become a serious safety hazard. The sources of pathogens in hospital sewage mainly include the following: patient excrement, waste from surgeries, wash water of the ward, and aerosol within the hospital. The bacteria include *Escherichia coli*, *Salmonella*, *Proteus*, *Shigella*, *Staphylococcus aureus*, *Clostridium botulinum*, *Clostridium difficile*, and so on. Viruses include norovirus, enteric adenovirus, rotavirus, cytomegalovirus, and so on. Typical parasites are *Microsporidium*, *Cryptosporidium*, *Tapeworm*, *Trematode*, *Hookworm*, and so on. The presence of these pathogens poses a potential threat to public health and the environment and must be controlled and removed through effective wastewater treatment measures [32,33]. This section mainly describes the main components of pathogenic microorganisms in medical sewage and the shortcomings of the current clinical use of surveillance methods.

### 2.1. Bacteria

#### 2.1.1. *Escherichia coli*

Enteropathogenic *E. coli* (EPEC) is a Gram-negative bacterium that poses a significant risk to human health, particularly in developing countries, by adhering to intestinal epithelial cells and causing diarrhoea, and is a significant cause of infant mortality [34]. The pathogenic mechanisms of EPEC include the formation of attaching and effacing (A/E) lesions that lead to structural destruction and functional changes in intestinal epithelial cells. Its associated diseases and complications mainly include diarrhoea, intestinal diseases, malnutrition, and growth disorders [35]. EPEC causes diarrhoea by producing heat-labile toxins (HLT) and/or heat-stable toxins (HST), especially in resource-poor areas [36]. Escherichia coli has long been used as an indirect measure of the public health risks associated with environmental water from faecal indicator bacteria (FIB). The monitoring of Escherichia coli in medical wastewater can be controlled and removed by effective sewage treatment measures [37,38]. Multiple-tube fermentation (MTF) is one of the standard methods to detect the presence of *E. coli* in water [39]. In this method, the samples to be tested are inoculated in a fermentation tube containing lactose peptone broth, and after the process of initial fermentation and re-fermentation, the nearest value of the number of coliform bacteria per litre of water can be calculated. The method procedure is multifarious, the cycle is long, it consumes a large amount of manpower material resources, and it is not suitable for the analysis of a large number of samples. However, the traditional method has the advantages of simple principle, low cost, and popularization.

#### 2.1.2. *Salmonella*

*Salmonella* is an important pathogenic bacterium with complex pathogenic mechanism and pathophysiological characteristics. Its pathogenic mechanism mainly involves various virulence factors, which function through different pathways. The virulence genes at Salmonella are mainly on *Salmonella* pathogenicity islands (SPIs), which play a key role in the invasion, survival, and spread of the strain outside the intestine [40].

Hospital sewage also contains high levels of antibiotics and drug-resistant bacteria. The widespread use of antibiotics has led to the presence of antibiotic-resistant bacteria in hospital wastewater, and the presence and spread of antibiotic-resistant *Salmonella* in wastewater has been repeatedly confirmed [41,42]. In a study from the Czech Republic [43], 89 strains of *Salmonella* were isolated from wastewater samples, 89% of which contained at least one drug-resistant strain, which was resistant primarily to sulphonamides and tetracycline. Wastewater treatment has played a role in reducing antibiotic resistant bacteria, but there is still a risk of residual and spread of resistant *Salmonella*. Therefore, the detection of *Salmonella* is very important to monitor the removal efficiency of pathogens in medical wastewater. *Salmonella* is usually detected by culture methods. After pretreatment of the sample vaccination to *Salmonella Shigella* agar (SS plate), it is sent to culture. *Salmonella* will form the centre under the SS plate with edge black, transparent colonies; these colonies can be further identified and detected through molecular biology. The total number of *Salmonella* can be detected by PCR amplification of the DNA fragment encoding the inv A gene. This method not only requires strict environment, but also needs an expensive test platform, and the need to design a suitable primer process is time-consuming, making it not suitable due to the lack of related equipment for rapid detection of pathogenic microorganisms in the sample [44].

#### 2.1.3. *Staphylococcus aureus*

Studies on antibiotic resistance in hospital wastewater have shown the presence of a variety of drug-resistant pathogens in wastewater [45], including methicillin-resistant *S. aureus* (MRSA). During wastewater treatment, these resistant strains can spread into the environment, posing a public health threat. MRSA and other drug-resistant strains in hospital wastewater have been reported in different countries and regions, indicating their widespread presence and diversity. Twelve *S. aureus* strains, all methicillin-sensitive *S. aureus* (MSSA), were isolated from 62 sewage samples at two wastewater treatment plants in Tunisia [46]. Another study also found that these strains included human- and animal-related genetic lineages. MRSA has also been detected in urban sewage treatment plants in Spain, indicating the presence of drug-resistant bacteria in urban sewage treatment plants [47]. MRSA survives in wastewater treatment plants and can become a reservoir for resistant bacteria, which poses a potential public health risk to wastewater treatment plant workers and people exposed to recycled water. Two major *S. aureus* clone complexes were found in the two wastewater treatment plants, one hospital-related and the other community-related [48]. These strains were still present during treatment and showed genetic diversity. Therefore, the monitoring of *staphylococcus aureus* in medical sewage can alleviate the pressure of public environmental pollution to a certain extent. The identification of *S. aureus* in clinical practice still requires the isolation of a single colony by culture and then identification by the latex agglutination method. Using the *S. aureus* identification kit, specific proteins are coated on the particles to prepare sensitization microspheres. When the specific sensitized microspheres are fused with the bacteria, the bacteria agglutinate with the specific protein on the surface of the sensitized microspheres. The rapid aggregation of latex particles into visible blue granular mixtures can finally identify *S. aureus*. Similarly, the identification method needs a certain culture, and it also has high false-positive rate, against which *S. aureus* fast detection still needs to be optimized.

#### 2.1.4. *Clostridium difficile*

*C. difficile* is a Gram-positive, sporogenetic, anaerobic bacterium that is widely distributed in the intestinal tract and environment of humans and animals. *C. difficile* is considered to be a major pathogen of antibiotic-associated diarrhoea and pseudomembranous colitis [49,50]. *C. difficile* is widespread in sewage treatment. Studies have shown that the pathogen is often present in raw sewage and is able to survive the sewage treatment process and then enter the wider environment through the treated sewage. *C. difficile* has been detected in several studies, including sewage treatment plants in southern Switzerland and Iran [51]. The diffusion of *C. difficile* in hospital wastewater system is mainly related to the biofilm formed on the inner wall of pipes. The shearing force of tap water may dislodge biofilm debris, which may colonize other parts of the wastewater system and facilitate the spread of *C. difficile* [52]. In addition, *C. difficile* can survive the wastewater treatment process, and strains are released into the environment, which raises the risk of possible transmission on wastewater reuse land [53]. Therefore, it is of great significance to monitor the content of *C. difficile* in medical wastewater; EIA and ELISA are the most commonly used immunodetection methods for toxins. These two methods directly detect toxins A and B in diarrheal stool samples by antigen-antibody reaction, are highly specific, simple to perform, and can provide test results within minutes to an hour [54,55]. But several studies have shown that the EIA diagnosis method has poor stability. The experimental results of Mohan et al. [56] showed that the sensitivity of EIAs was between 50% and 90%. The reason for the unstable sensitivity may be due to a variety of factors such as antigenic variation of toxins from different strains, the influence of antigen concentration in wastewater, and differences in operating techniques in different laboratories.

### 2.2. Viruses

#### 2.2.1. Enterovirus

Enterovirus is an important pathogen that causes central nervous system (CNS) infection and often leads to various nervous system diseases [57]. These viruses belong to the Picornaviridae family, which includes poliovirus, rhinovirus, enterovirus A71, and enterovirus D68. The epidemiology and surveillance of viruses in wastewater is an important area of public health research, and studies have shown that enteroviruses in wastewater can be used as an indicator of the prevalence of related viral infections. A large-scale survey conducted in a metropolitan area in southern Italy showed that wastewater treatment plants present certain challenges in eliminating enteroviruses [58]. The wastewater-based epidemiology (WBE) has been used in several studies to track the transmission dynamics of the virus. For example, in a study conducted in the city of Milan, by monitoring enteroviruses (EV-RNA) in wastewater, researchers found that the viral trend in wastewater was from eight to five weeks earlier than in clinical samples [59], which can serve as an early warning of an outbreak. There are many ways to detect enterovirus, and traditional detection methods such as the cell culture method, neutralization test, and immunofluorescence method are complicated and time-consuming. Clinical commonly used nucleic acid detection methods need precise instruments and professional technical personnel. Immunological detection methods, especially the immunogold labelling method and enzyme-linked immunosorbent assay, have been favoured for their advantages of rapid detection, but they still have shortcomings such as inaccurate quantification [60]. At present, electrochemical detection based on immunology has high sensitivity and can easily, quickly, and quantitatively detect intestinal virus. For the clinical detection of intestinal viruses, this provides a new way of thinking.

#### 2.2.2. Respiratory Viruses

Respiratory viruses monitored in medical wastewater include SARS-CoV-2, Influenza A, Respiratory syncytial virus (RSV), and Measles virus [61]. Studies have shown that monitoring RSV concentrations in medical wastewater can predict respiratory syncytial virus hospitalizations (RSVH) surges in children up to 12 days in advance. Similarly, research in Spain has shown that viral data in wastewater can reveal that the actual number of cases has been underestimated, and that asymptomatic infections may be more frequent than expected. Rapid, real-time wastewater surveillance is a non-invasive, cost-effective, and early warning epidemiological tool that can be used to monitor respiratory virus transmission in communities, contributing to public health system preparedness and response. Rapid field antigenologic diagnostic tests for Influenza A, RSV, and SARS-CoV-2 are readily available but are not quantitative and are less sensitive than laboratory tests. Wastewater monitoring for epidemiological warning needs to be highly sensitive to reveal the trend of the spread of the virus. Multiplex detection based on PCR is available in many clinical laboratories. This method and cell culture are the main means of epidemiological surveillance at present, but the disadvantage is that the monitoring time is long, which is not conducive to the rapid response of the health system.

#### 2.2.3. Hepatitis Virus

In the Tunis study, the molecular detection and genetic characterization of hepatitis A virus (HAV) showed that the spread of HAV in the environment was closely related to socioeconomic levels and sanitary conditions, existing wastewater treatment processes could not effectively remove the virus, and treatment systems needed to be improved [62]. In South Africa, the unique HAV IB subtype has been detected in wastewater and surface water, and some of these strains had amino acid mutations on neutralizing epitopes, showing variability [63]. Real-time monitoring of hepatitis virus content in medical wastewater can prompt wastewater treatment centres to improve wastewater treatment processes, so as to effectively reduce the environmental transmission of the virus and reduce the risk of infection. LAMP is a rapid, simple, highly sensitive nucleic acid amplification techniques that can be used to detect DNA or RNA viruses. LAMP technology amplification targets virus specific DNA sequences and generates real-time visual signals, thus realizing the rapid and sensitive to the virus detection. For hepatitis c virus (HCV), LAMP has been used to distinguish and detect the different genotypes, such as HCV—1 b, 2 a, 3, and 6 a [64]. The LAMP test showed high specificity and sensitivity and a LOQ range of 1.0 × 10^3^ to 1.5 × 10^3^ IU/mL. LAMP can be used to test the performance of the virus in sewage directly and has been assessed in the study of multiple viruses [65]. LAMP has emerged as a powerful diagnostic tool for rapid detection. However, in complex samples such as sewage, its detection sensitivity may still be affected, especially in the process of sample processing and DNA extraction, there are many error-prone links. But at the same time, it also has complex primer design, high false-positive rate, and expensive raw materials.

### 2.3. Parasites

The problem of parasites in medical wastewater is of great concern. Hospitals are one of the main sources of intestinal parasites in municipal wastewater, including protozoa and vermicular eggs [66]. In studies in different regions, such as wastewater treatment plants in Iran [67], Morocco [68], and India [69], a variety of parasites has been detected, and their treatment efficiency is different. Some studies have also evaluated the parasite removal effectiveness of different wastewater treatment systems and found significant differences in treatment efficiency. Studies have shown that parasites in hospital wastewater may still exist in treated wastewater after entering urban wastewater treatment systems, affecting farmland irrigation and human health downstream [8]. Therefore, monitoring and improving the efficiency of parasite removal in wastewater treatment systems is a key issue. The etiological examination of parasites refers to the collection of wastewater samples, direct observation through microscopy, or observation after treatment and staining, in order to find parasitic worms or eggs as the basis; sometimes, there is a need to carry out parasite culture in vitro or animal vaccination and then carry out etiological examination. Subsequent molecular biological tests can only be performed after the etiological examination isolates and confirms the presence of parasites. The inspection process is time-consuming, laborious, and has poor repeatability, and the results are completely judged by the inspector, who needs rich parasitological knowledge and detection experience. Therefore, this scheme is not suitable for large-scale sewage monitoring.

### 2.4. Fungi

The severity of fungal infections depends on the site of infection and host immunity. Among them, *Candida albicans* can cause a variety of diseases, from mucosal infection to systemic infection in people with low immunity [70,71]. *C. albicans* is capable of asymptomatic colonization in a variety of sites in the host body, especially in the gastrointestinal and urogenital tracts. In one study, floating cells and biofilm cells of *C. albicans* and other non- *C. albicans* were isolated from hospital wastewater, suggesting that untreated medical wastewater may potentially cause fungal disease in immunocompromised populations [72,73]. Traditional laboratory diagnostic methods for fungi include direct microscopic examination, culture, and identification, and clinically, flight mass spectrometry is usually used to identify the cultured flora. The problem of this method is that the equipment is expensive, it is limited to the fungal protein database of the flight mass spectrometer, and there is a large error. So far, the traditional mycological detection method is still irreplaceable in clinical practice.

Based on above discussion, inspection methods of main pathogens of medical wastewater and related diseases are listed in Table 1.

## 3. Electrochemical Biosensors for Detecting Pathogens in Medical Wastewater

Electrochemical sensors are generally composed of transducers and recognition elements, which are specifically combined with the target detection of pathogens and microorganisms, causing changes in the impedance and current in the electrochemical system and detecting the concentration of pathogenic microorganisms through the measurement of electrical signals [86,87,88]. According to the input and output of electrical signals, electrochemical biosensors can be divided into voltammetry, conductance, impedance, photochemistry, and electrochemical luminescence. Table 2 summarizes the indicators of the current rapid detection methods using electrochemical biosensors to detect pathogenic microorganisms. By designing different recognition elements, biosensors can detect different microorganisms. By combining the immunological principles and molecular biology principles widely used in clinical practice, they can be widely used in the detection of various pathogenic microorganisms. Therefore, biosensors are expected to be further studied and popularized in the detection of pathogenic microorganisms.

### 3.1. Volt-Ampere Type Biosensor

Voltammetry biosensors assess the presence and concentration of bacteria and contaminants by measuring the current change at the applied potential, and the strength of the current has a certain linear relationship with the concentration of the substance to be measured [89,90]. In electrochemical sensors, voltammetry is a commonly used technique, including cyclic voltammetry, square wave voltammetry, and stripping voltammetry. For example, in bacterial detection, voltammetry biosensors can quickly and sensitively identify and quantify bacterial species, such as the label-free amperometric biosensor for *E. coli* O157:H7 [91]. A previous study proposed an amperometric biosensor platform for the detection of pathogenic *E. coli* O157. The biosensor was based on the immunological principle of antibody-antigen interaction, in which antibodies were covalently bound to the surface of a nickel oxide (NiO) thin film substrate by sputtering technology as show in Figure 1. The NiO substrate accelerated charge transfer and provided excellent support for antibodies, which improved target acquisition efficiency and enabled the direct detection of *E. coli* O157 without labels. Electrochemical detection results showed that the sensor had a wide linear range of 10^1^–10^7^ cells/mL and was highly selective and specific to other bacterial species. In addition, electrochemical biosensors usually modify marker materials on recognition elements to amplify detection signals, thus realizing low abundance detection. For example, the principle of the interaction between nano-anti-schistosomiasis conjugate and Soluble Egg Antigen (SEA) was studied, and the structure of the nano-biosensor was characterized by rabbit anti-schistosomiasis antibody as show in Figure 2. In the presence of antibodies, the gold nanoparticles (AuNPs) strips were oxidized to form active gold oxides with electrons, which produced redox-reduction signals, and then analysed by cyclic voltammetry. Studies showed that the detection concentration of schistosomiasis antigen in faeces ranged from 1.13 × 10^1^ ng/mL to 2.3 × 10^3^ ng/mL. The results showed that the test strip could be used to detect schistosomiasis antigen in real samples [92]. Recently, a team has researched and developed an electrochemical biosensor that combined side-flow technology and electrochemical technology to detect *E. coli* O157 [93]. The electrodes were installed below the lateral flow detection line, while organic–inorganic nanoparticles prepared with *E. coli* O157-specific antibodies were attached to the lateral flow detection line. When *E. coli* O157 was detected, the organic–inorganic nanoparticle and *E. coli* O157 antimicrobial peptide-labelled ferrocene sandwich structures were formed on the lateral flow detection line. The team developed a portable current detection device and used a smartphone-based device to detect current signals. The electrochemical biosensor could specifically detect *E. coli* O157 with an LOD of 25 CFU/mL.

### 3.2. Conductive Biosensor

In recent years, reports on conductive biosensors have been increasing. Its principle is to determine the concentration of the target detection object by measuring the change of the conductance difference between two parallel electrodes [94,95,96]. A rapidly innovative conductive biosensor utilizing a polypropylene microfiber membrane coated with conductive polypyrrole and functionalized antibodies for the field detection of *E. coli* O157 has been demonstrated [97]. The pathogen-specific antibody was covalently attached to the conductive membrane by glutaraldehyde and then sealed with 5% bovine serum albumin solution. When these membranes were exposed to *E. coli* O157 in a phosphate buffer, the resistance on the electrode surface increased when a voltage was applied, indicating the presence of the pathogen. The biosensor system was shown to be able to distinguish small changes in conductivity due to the presence of the target pathogen in solution over a sensitivity range of 1 × 10^−2^~1 × 10^−1^ cells/L. A research team has prepared a biosensor based on conductive polymer nanowires for monitoring bacterial spores [98]. By assembling polypyrrole nanowires (PPY) on microfabricated gold cross-fingered microelectrodes, they constructed a chemically resistant biosensor that could be used for monitoring *Bacillus globigii*. The biosensor had a good linear correlation (r^2^ = 0.992) in the concentration range of 1–100 CFU/mL and a response time of 30 min, which could be used for rapid detection of medical wastewater.

### 3.3. Impedance Biosensor

Electrochemical impedance spectroscopy (EIS) is used to measure impedance by applying a small AC voltage to the sensor electrode over a certain frequency range and measuring the resulting AC current [99,100,101]. The impedance includes two main parts, resistance and capacitance, and the combined event will change the charge distribution and mass transfer process on the electrode surface, thus changing the resistance and capacitance of the system [102]. By analysing the impedance spectrum (such as the Nyquist diagram or Bode diagram), the degree of binding to the target molecule can be determined. By comparing the impedance changes before and after binding, the concentration of the target molecule can be quantitatively analysed, and the impedance change is usually proportional to the concentration of the target molecule, so a calibration curve can be established for the quantitative detection of the actual sample. The impedance biosensor used for the detection of pathogenic microorganisms mainly combines the principle of immunology. A research team has used this principle to develop an electrochemical method for the rapid detection of *Salmonella typhi*, in order to capture antibody-modified magnetic nanoparticles (MNPs), enzymic probes for the detection of antibody-modified MNPs, and the main raw materials of microfluidic chips, and mixed them to form a composite [103]. High-impedance glucose was then injected into the chip to generate high-impedance hydrogen peroxide and low-impedance gluconic acid, which was finally measured using low-cost cross-finger microelectrodes and an electrochemical impedance analyser to identify the target bacteria. Under the best conditions, the detection range of the biosensor was 1.6 × 10^2^~1.6 × 10^6^ CFU/mL concentration of *Salmonella typhi*, and the LOD was 73 CFU/mL. A research team [104] prepared a self-assembled lectins (ricin A and ricin lectins) monolayer of electrical impedance biosensors on polyelectrolyte-modified electrodes by electrostatic interaction for bacterial detection. Target microorganisms were captured by the detection interface through the interaction of lectins such as ricin A and glyco-based complexes on the cell surface. The combination of microorganisms and lectins increased the electron transfer resistance of the electrode, and the types of pathogenic microorganisms were identified according to a series of linear relationships. The sensor had high sensitivity in the selective identification of *E. coli* DH5a, *Enterobacter cloacae*, *Bacillus subtilis*, and other strains.

### 3.4. Photoelectric Chemical Biosensor

The photoelectric chemical (PEC) sensor is a kind of sensor technology based on electrode/interface photoinduced electron transfer process. Due to its high sensitivity, high anti-interference, and high-cost performance, it has become more and more popular in recent years. The PEC biosensor is a kind of biological detection technology based on photoelectric chemical reaction. The basic principle is that when light shines on the photosensitive material, the light energy is converted into electrical energy, which triggers the photoelectric chemical reaction, so as to achieve the detection of biological targets. Some teams [105] selected BiOI semiconductor films as photoelectric conversion units and visible light drive (VLD) photocatalytic antibacterial units, AuNPs as linking units, and 4-mercaptophenylboric acid (4-MPBA) as antibiotic-free capture units. Based on the reversible binding of boric acid groups to peptidoglycan on the bacterial cell wall, the integrated photoelectrochemical PEC platform had excellent capture performance and a high detection sensitivity for *E. coli*, with an LOD of 46 CFU/mL. Photochemical strategies can also use conjugated polymers to distinguish microbial pathogens, and a team [106] has used cationic poly derivatives (PPVs) as photochemical biosensors, as shown in Figure 3. PPVs have excellent photoelectric conversion properties and can bind to pathogens through electrostatic and hydrophobic interactions. PPV was excited under 450 nm light irradiation and transferred electrons to oxygen in the electrolyte, producing a strong photocurrent signal. After the introduction of pathogenic bacteria, steric hindrance was generated to block electron transfer, and a signal-off type PEC sensor was proposed. The different size of pathogenic bacteria showed different resistances of electron transfer, which resulted in different degrees of decreases of the photocurrent. Based on this principle, pathogenic microorganisms could be easily and sensitively identified. The sensor could detect different concentrations of pathogenic microorganisms in the range of OD_600_ = 0.5 to 2 and showed the linear relationship between the photocurrent density and the concentration of pathogenic microorganisms, according to the principle of simple and sensitive identification of pathogenic microorganisms. A new label-free PEC immune sensor platform based on rare earth doping and quantum dots (QDs) sensitization scheme has been developed for *Vibrio parahemolyticus* (VP) detection [107]. The doping of CdSe nanoparticles improved the absorption of TiO_2_ in the visible region and enhanced the photocurrent response of PEC sensors. This PEC immune sensor had an LOD of 25 CFU/mL and a wide detection range of 10^2^~10^8^ CFU/mL, with high stability, low price, and a short detection time. The sensor could be used to detect other pathogenic microorganisms by replacing antibodies, so the sensor is expected to be one of the fast and ultra-sensitive detection of pathogenic microorganisms in medical wastewater.

### 3.5. Electrochemical Luminescent Biosensor

The electrochemical luminescence (ECL) biosensor has significant advantages in pathogen detection [108,109]. Traditional techniques such as PCR and ELISA, while accurate and effective, are limited due to their complexity and are time-consuming [110,111,112]. ECL biosensors combine the advantages of electrochemical and photoluminescence analysis for high sensitivity and the simple detection of pathogens. A research team [113] has established an ECL biosensor based on the relationship between resonant energy transfer and surface plasma coupling. They effectively quenched the ECL signal of quantum dots due to resonance energy transfer according to nanomaterials. When identifying the target, the DNA bound by the nanomaterial changed from hairpin structure to linear conformation, the distance between BN quantum dots and AuNPs increased, and the surface plasma coupling effect enhanced the ECL signal by about six times. The sensor could accurately quantify the Shiga toxin-producing *E. coli* (STEC) gene with LOD of 0.3 pmol/L. In recent years, ECL sensors have made significant progress in the detection of pathogens, especially in the development of paper-based ECL platforms, which show good promise in the field diagnosis of pathogens [114]. In addition, a recent study prepared an ECL immune sensor for mycotoxin detection using Zr-MOF nanoplates and Au@MoS nanoflower with a detection range from 0.0001 to 100 ng/mL and LOD of 0.034 pg/mL [115]. The sensor has been successfully applied in many practical samples, which provides a new idea for theindirect detection of pathogenic microorganisms. The specific electrochemical detection of pathogenic microorganisms described in this section can be seen in Table 2.

**Table 2 molecules-29-03534-t002:** Comparison of detection methods for pathogenic microorganisms.

Detection Method	Target Detector	LOD	References
Volt-ampere type biosensor	*E. coli* O157	2.5 × 10^−2^ CFU/L	[91]
Volt-ampere type biosensor	*Schistosomiasis* antigen	1.13 × 10^1^~2.3 × 10^3^ ng/mL	[92]
Conductive biosensor	*E. coli* O157	1 × 10^−2^~1 × 10^−1^ cells/L	[97]
Conductive biosensor	*Bacillus anthracis*	1 × 10^−3^ CFU/L	[98]
Impedance biosensor	*Salmonella typhi*	7.3 × 10^−2^ CFU/L	[103]
Impedance biosensor	*E. coli* DH5a, *Enterobacter cloacae*, *B. subtilis*, *Saccharomycetes*	high sensitivity	[104]
Photoelectric chemical biosensor	*E. coli*	4.6 × 10^1^ CFU/L	[105]
Photoelectric chemical biosensor	*Vibrio parahemolyticus*	2.5 × 10^1^ CFU/L	[106]
Electrochemical luminescent biosensor	Shiga toxin *E. coli*	0.3 pmol/L	[113]
Electrochemical luminescent biosensor	Mycotoxin	0.034 pg/mL	[115]

## 4. Discussion and Prospect

For sewers or sewage treatment plants in medical areas, sewage monitoring can provide early warning for disease outbreaks in a region or city [116]. Another important role of sewage surveillance is that it can detect the variation of virus strains through phylogenetic analysis, which provides valuable information for tracking the variation of the virus over time and different regions. Sewage monitoring can also trace levels of viruses in the wastewater seasonal fluctuations, and epidemiological patterns reflect the virus in the community. Sewage monitoring can be used as an early indicator that an infected population is increasing or decreasing. Measuring the effect of public health interventions, researchers can see whether sewage virus signals increase or decrease after interventions such as vaccination or school closures, and they can monitor whether cases are rising or falling in larger populations in cities by sampling sewage treatment plants. Sewage surveillance can create an early warning response mechanism for future pandemic prediction, and it is an important source of information on the health status of urban populations and the development trend of infectious diseases. In the future, it is believed that the robust health surveillance system based on sewage monitoring will play an increasingly important role as an important weapon for the effective and timely prevention and control of emerging, seasonal, and even future infectious diseases.

For the detection of pathogenic microorganisms in medical wastewater, the traditional culture detection method is still a commonly used and accurate and effective detection method in clinical practice [117]. However, this method takes a long time to conduct the microbial culture and has many operation steps, which has difficulty in meeting the needs of rapid detection. The introduction of pathogenic microorganisms in medical wastewater at the same time has led to the nucleic acid test and immunology test as new types of tests in clinical practice. During the COVID-19 epidemic, RT-PCR-based nucleic acid detection methods were fully applied in the biomedical field [118]. However, nucleic acid detection methods or immunological detection methods need to be analysed in the laboratory, which has difficulty meeting the on-site detection needs of medical wastewater. This article focuses on electrochemical analysis based on the principle of biological sensors, which can specific detection target pathogenic microorganisms, and at the same time, has high sensitivity and low cost, a simple process and no professional operation, with the advantage of on-site rapid detection. An electrochemical biosensor based on the technology of lateral flow [93], for example, by replacing the antibodies in the organic–inorganic nanoparticles, can be used to identify other target pathogenic microorganisms. In addition, the manufacturing process of electrochemical biosensors is relatively simple and does not require complex micromachining techniques or expensive equipment. For example, printed electrodes [119] and drip coating techniques [120] can be used to manufacture these sensors, and these processes are simple and inexpensive. Electrochemical biosensors can therefore be mass-produced through large-scale manufacturing processes, which greatly reduces the cost of individual sensors. Mass production can also lead to increased production efficiency and further reduction in unit costs. At the same time, such sensors can also have the potential to detect other pathogens in biomedical research and clinical diagnosis. A biosensor for detecting pathogenic microorganisms of medical wastewater is still at the primary stage of study, and there is no uniform standard. Therefore, a variety of different sensor design structures leads to a huge performance difference. How to improve the sensor’s repeatability, stability, and anti-interference ability, as well as make it easily applicable in the actual detection, are problems that must be solved. In addition, the biosensor often has certain restrictions on the target analysis of the concentration of the sample request in medical wastewater samples of pathogenic microorganisms and enrichment methods, and this is one of the problems in urgent need of research [121].

Since the application of electrochemical hybridization probe DNA technology in 2006 [122], the field of electrochemical biosensors has experienced significant progress, and the development of electrochemical hybridization probe DNA sensors has further improved the level of unlabelled nucleic acid detection. In 2008 [123], carbon nanotubes were combined with electrochemical biosensors for the detection of microbial metabolites. In 2010 [124], a team combined microfluidic chips with electrochemical biosensors for the rapid detection of bacteria and viruses. Microfluidic technology enables automated sample handling and analysis and improves the detection efficiency. In 2012 [125], the introduction of two-dimensional materials improved the sensitivity of electrochemical biosensors. In 2014 [126], the development of nanotube-antibody-DNAzyme composites led to the construction of highly sensitive immunosensors for pathogen detection. In 2016 [127], through a portable electrochemical biosensor combined with multichannel optical biosensor and a smartphone, real-time data transmission and remote monitoring through wireless communication technology could be used for large-scale environmental microbial monitoring. In 2020 [128], artificial intelligence (AI) and machine learning algorithms were used for data processing and analysis, improving the accuracy and efficiency of microbial detection. In 2023 [129], the development of nanozymes and biomimetic materials was widely used in electrochemical biosensors, which further improved the stability and detection sensitivity of sensors, marking another major breakthrough in electrochemical biosensor technology. The specific flow chart is shown in Figure 4. The current electrochemical biosensor design has a flexible structure, mainly through its combination with the excellent performance of nano functional materials [130] and sophisticated microfluidic chip technology [131]; this kind of sensor shows a high detection capability. At the same time, the rapid development of AI learning in recent years has brought beneficial changes in many fields, including the field of the environment. Internet-based electrochemical biosensors can conduct AI learning through big data statistics and other methods, as well as analyse medical wastewater to predict the current epidemic trend in the region [132], which has great research value and application potential.

## 5. Conclusions

Medical wastewater may do great harm to public health and the ecological environment if it is not discharged and disinfected properly due to its high content of pathogenic microorganisms. There is an urgent need for the rapid detection of pathogenic microorganisms in medical wastewater. Electrochemical biosensor technology is widely used in the detection of pathogenic microorganisms in medical wastewater because of its simple, fast, and real-time monitoring. With the introduction of highly specific and sensitive recognition elements and the development of various signal transduction technologies, various electrochemical biosensors have realized the trace analysis of microorganisms. In the past few years, the rapid development of biosensors has met the needs of the market and society, providing an effective solution for the detection of pathogenic microorganisms in medical wastewater. Future research should continue to optimize sensor sensitivity and specificity and develop more portable, low-cost sensor devices to meet the needs of different application scenarios. In addition, rapid detection methods for different types of pathogenic microorganisms should be further explored and innovated to provide more powerful technical support for public health and environmental protection.

## Figures and Tables

**Figure 1 molecules-29-03534-f001:**
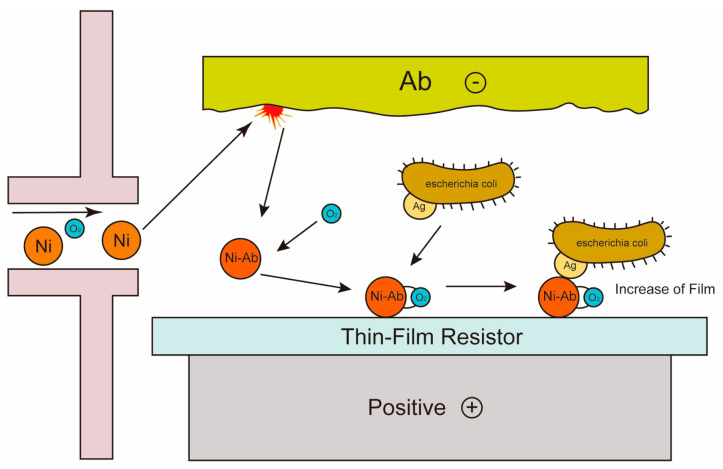
The label-free amperometric biosensor for *E. coli* O157:H7.

**Figure 2 molecules-29-03534-f002:**
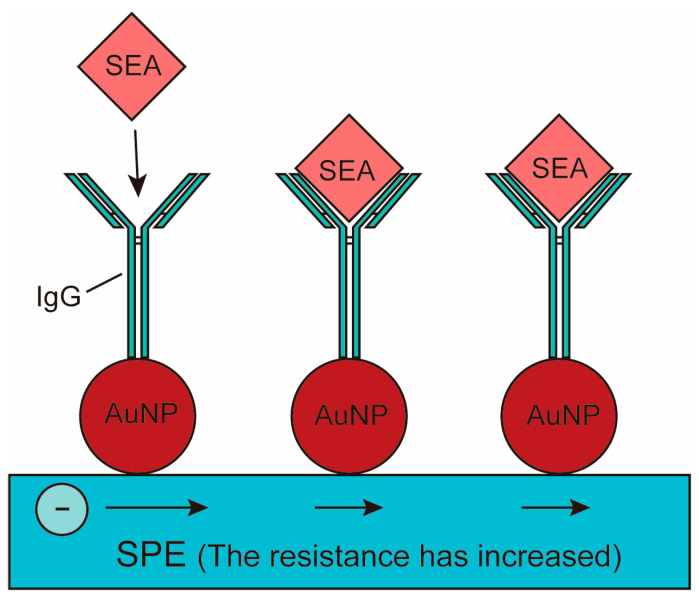
Voltametric biosensor capable of detecting schistosomiasis antigens in faecal samples.

**Figure 3 molecules-29-03534-f003:**
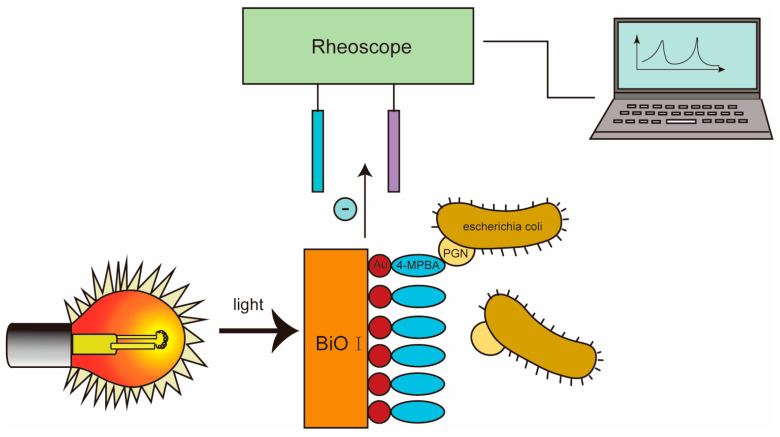
Integrated photoelectrochemical platform for antibacterial detection with BiOI semiconductor films and gold nanoparticles.

**Figure 4 molecules-29-03534-f004:**
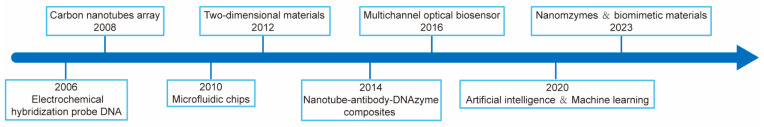
The development of electrochemical biosensors.

**Table 1 molecules-29-03534-t001:** Inspection methods of main pathogens of medical wastewater and related diseases.

Pathogen	Source	Disease	Detection Type	References
**Bacteria**				
*E. coli*	Faeces of patients	Gastroenteritis	MTF LOQ: 1 × 10^3^ CFU/L	[74]
*Salmonella*	Faeces of patients	Salmonellosis, bacteremia	RT-PCR LOD: 7.5 × 10^−2^ CFU/L	[75]
*S. aureus*	Tissue fragments	Gastroenteritis, bromatoxism	Immunofluorescence technique LOD: 1 × 10^−2^ CFU/L	[76]
*C. difficile*	Faeces of patients	Pseudomembranous colitis, antibiotic-associated diarrhoea	EIASA Sensitivity: 69%~99%	[77]
**Virus**				
Adenovirus	Aerosol	Respiratory tract infection	RT-PCR LOQ: 1.5 × 10^1^ copies/mL	[78]
HAV	Faeces of patients	Infectious hepatitis	RT-PCR LOD_95_: 8.02 × 10^2^~2.80 × 10^3^ copies/mL	[79]
Norovirus	Faeces of patients	Gastroenteritis	RT-PCR LOD_95_: 8.02 × 10^2^~2.80 × 10^3^ copies/mL	[79]
SARS-CoV-2	Aerosol	Respiratory infections	RT-PCR LOD: 5 × 10^1^ copies/mL	[80]
**Fungi**				
*C. albicans*	Wash water of the ward	Candidemia	Microfluidics LOD: 1.2 × 10^−3^ CFU/L	[81]
*Rhizopus*	Wash water of the ward	Mucormycosis	LAMP LOD: 1 × 10^−1^ pg/μL	[82]
*Aspergillus fumigatus*	Aerosol	Aspergillosis	RT-PCR LOD: 40 fg; LOQ: 400 fg	[83]
**Parasite**				
*Cryptosporidium*	Faeces of patients	Severe chronic diarrhoea	LAMP LOD: 1 × 10^−2^~1 × 10^−3^ pg/μL	[84]
*Tapeworm*	Faeces of patients	Cestodiasis	LAMP LOD: 1 × 10^−2^~1 × 10^−3^ pg/μL	[84]
*Ascaris* spp.	Faeces of patients	Ascariasis	Microscopy LOD: 50 seeded eggs/gr faeces	[85]

LOD: limit of detection; LOQ: limit of quantitation.

## Data Availability

The original contributions presented in the study are included in the article, further inquiries can be directed to the corresponding authors.

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
