# Peer review of "Research Progress on Detection of Pathogens in Medical Wastewater by Electrochemical Biosensors"

_molecules, 2024, doi:10.3390/molecules29153534_

Round 1

Reviewer 1 Report

Comments and Suggestions for Authors

Authors should revise the manuscript.

Comments on the Quality of English Language

Author Response

Responds to the reviewer’s comments:

Reviewer #1:

Electrochemical biosensors have emerged as a promising technology for detecting pathogens in medical wastewater due to their high sensitivity, specificity, rapid response, and potential for on-site monitoring. Research in this area has been actively progressing, focusing on various aspects such as sensor design, biorecognition elements, signal transduction mechanisms, and practical applications. The present review aims to provide a brief discussion on the role of electrochemical biosensors in detection of pathogens in medical wastewater. Review is too generic and does not offer any new insights. Authors should revise the manuscript.

Response: We are sincerely grateful for the reviewer’s positive and useful comments to our manuscript. We carefully revised the paper in detail based on the reviewer’s comments or suggestions.

Major Comments

  1. Comment: Introduction should also include the statistical data about the presence of various pathogens in biomedical wastewater as literature is available.

Response: The authors are grateful for the reviewer’s valuable suggestion. We have added some statistics on the presence of various pathogens in the introduction section. For example, an experimental team used 16S rRNA gene fingerprinting to estimate the abundance of the total population of potential pathogenic bacteria in Hong Kong municipal wastewater, which was found to be 0.06-3.20%.

  1. Comment: Section 2 is generic and does not offer any discussion based on several systematic reviews and research articles relevant to the topic.

Response: We are sincerely grateful for the reviewer’s positive and useful comments to our manuscript. In the revised manuscript, we have added common detection methods for pathogenic microorganisms related to the subject in each sub-section of the Section 2, and added corresponding evaluations.

  1. Comment: Authors, can tabulate the research articles, highlighting the pathogens, source, adverse effects, and present detection methods (LOD and LOQ as well).

Response: We thank the reviewer’s precious and professional suggestion. According to reviewer’s request, we have inserted the relevant table (Table 1) at the end of the Section 2 of the article for readers' reference.

  1. Comment: Section 3 is brief does not offer any recent research progress discussion.

Response: We thank the reviewer’s precious and professional suggestion. We have added some cases study of recent electrochemical biosensor research in recent years to the Section 3.

  1. Comment: Authors can tabulate the findings of research articles relevant to the sub-section.

Response: The authors are grateful for the reviewer’s professional comment. According to reviewer’s request, we inserted the relevant table(Table 2) in the Section 3 of the article.

  1. Comment: Authors can also tabulate the advantages and disadvantages of the stated biosensors.

Response: The authors are grateful for the reviewer’s valuable suggestion. The authors have added the corresponding advantages and disadvantages of the stated biosensors in second paragraph in a new Section 4 (Section of Discussion and Prospect).

  1. Comment: Should include the economic aspects of the biosensors compared to available detection methods.

Response: The authors are grateful for the reviewer’s valuable suggestion. The authors have added the corresponding economic aspects of the biosensors in Section 4.

  1. Comment: Authors should also include the future prospects and limitations of the biosensors based detection of pathogens.

Response: We thank the reviewer’s precious and professional suggestion. The authors have added a fourth section "Discussion and Prospect" to explore the prospects and corresponding limitations of pathogenic microbial testing for medical wastewater.

Minor Comments

  1. Comment: Line 20, 83: Scientific names in italics.

Response: The authors are grateful for the reviewer’s valuable suggestion. We have made correction according to the Reviewer’s comments.

  1. Comment: Line 91, 139: Gram-negative. Gram-positive. Gram, first letter capital.

Response: We thank the reviewer’s precious and professional question. We have carefully checked and proofread the articles for spelling.

  1. Comment: Line 99, 101: E. coli.

Response: The authors are grateful for the reviewer’s valuable suggestion. We have made correction according to the Reviewer’s comments.

  1. Comment: Scientific names of microorganisms should be stated in full only at first mention, later on Genus should be abbreviated, ex E. coli. Check the same in entire manuscript.

Response: The authors are grateful for the reviewer’s valuable suggestion. We have carefully checked the use of scientific names of microorganisms in the manuscript.

  1. Comment: Line 213 to 219: Remove bold.

Response: The authors are grateful for the reviewer’s valuable suggestion. We have removed bold.

  1. Comment: References should be journal format.

Response: The authors are grateful for the reviewer’s valuable suggestion. Our references section has been changed to Molecules’s format.

Reviewer 2 Report

Comments and Suggestions for Authors

The review paper presented entitled "Research progress on detection of pathogens in medical wastewater by electrochemical biosensors" do not show the progress of the electrochemical technology since first, do not establish a clear period of the research progress, including studies from 2006 to 2023. Thus, some reference are very old and a lot of progress was accomplished during such a vast period. Moreover, the presented paper is very poor in terms of content. First, in the introduction are only mentioned electrochemical biosensor as mainly detection methods described, where there are many others that as least should be mention in the introduction of the paper. More, in introduction, authors state that medical equipment as wells as chemicals and pharmaceutical waste is discharge in sewage??? This should certainly be revised. Another point to mention is the unwritten of the bacteria specie, that should always be written in italic. 

Generally, the revision is very poor, a lot more references should be included in the text, mainly devoted to the description of electrochemical technologies and respective applications. Within the description of the technologies a table (or several) should be included with main features of the sensors described in the text. Also, is not clear why authors was chosen these references and not others.  Authors should should state in text main differences between technologies presented and main advantages. 

Therefore, I believe the manuscript is not ready for publication at this stage.

Comments on the Quality of English Language

the quality of English and written language is suitable

Author Response

Responds to the reviewer’s comments:

  1. Comment: Do not show the progress of the electrochemical technology since first, do not establish a clear period of the research progress, including studies from 2006 to 2023.

Response: We are sincerely thanks for the reviewer’s positive and useful comments to improve the quality of manuscript. We have added content on the electrochemical technology application in biosensor since 2006. The figure about the development history was added in this version as show in Figure R1. The discussion about development was provided in the manuscript as well.

Figure R1 The development of electrochemical biosensors

  1. Comment: In the introduction are only mentioned electrochemical biosensor as mainly detection methods described, where there are many others that as least should be mention in the introduction of the paper.

Response: We are sincerely grateful for the reviewer’s positive and useful comments to our manuscript. We have added the corresponding techniques in the introduction section. For example: polymerase chain reaction (PCR) and loop - mediated isothermal amplification (LAMP), rapid immunological assays such as enzyme immunoassay (EIA) and enzyme linked immunosorbent assay (EIASA) .

  1. Comment: In introduction, authors state that medical equipment as wells as chemicals and pharmaceutical waste is discharge in sewage??? This should certainly be revised.

Response: Thanks for your carefully review. We have revised this sentence as following:

The sources of pathogens in hospital sewage mainly include the following: patient excrement, waste from surgeries, wash water of the ward, and aerosol within the hospital.

  1. Comment: Another point to mention is the unwritten of the bacteria specie, that should always be written in italic.

Response: The authors are grateful for the reviewer’s valuable suggestion. We have carefully checked the use of scientific names of microorganisms in the manuscript.

  1. Comment: A lot more references should be included in the text, mainly devoted to the description of electrochemical technologies and respective applications.

Response: We thank the reviewer’s precious and professional suggestion. In this manuscript, the Section 4 "Discussion and Prospect" is added to describe the detection methods of microorganisms in medical wastewater and the application prospect of electrochemical technology.

  1. Comment: Within the description of the technologies a table (or several) should be included with main features of the sensors described in the text.

Response: The authors are grateful for the reviewer’s professional comment. According to reviewer’s request, we inserted the relevant table (Table 2) in the Section 3 of the article to present main features of the sensors described in the text.

  1. Comment: Authors should state in text main differences between technologies presented and main advantages.

Response: We are sincerely grateful for the reviewer’s positive and useful comments to our manuscript. In the revised manuscript, we have added common detection methods for pathogenic microorganisms related to the sub-section in each section of the Section 2, and added corresponding discussion on advantages of technologies the Section 4.
